



# Burned Area and Carbon Emissions Across Northwestern Boreal North America from 2001-2019

Stefano Potter[1], Sol Cooperdock[1,2], Sander Veraverbeke[3], Xanthe Walker[4], Michelle C. Mack[4],

Scott J. Goetz[5], Jennifer Baltzer[6], Laura Bourgeau-Chavez[7], Arden Burrell[1], Catherine Dieleman[8]

Nancy French[7], Stijn Hantson[9], Elizabeth E. Hoy[10], Liza Jenkins[7], Jill F. Johnstone[11], Evan S.

Kane[12], Susan M. Natali[1], James T. Randerson[13], Merritt R. Turetsky[14], Ellen Whitman[15],

Elizabeth Wiggins[16], and Brendan M. Rogers[1]

[1]Woodwell Climate Research Center, Falmouth, MA, 02540 USA
[2]University of California, Los Angeles, CA, 90095 USA
[3]Faculty of Science, Vrije Universiteit Amsterdam, Amsterdam, The Netherlands
[4]Center for Ecosystem Science and Society, Northern Arizona University, Flagstaff, AZ, 86011 USA
[5]School of Informatics, Computing, and Cyber Systems, Northern Arizona University, Flagstaff, AZ, 86011 USA
[6]Wilfrid Laurier University, Waterloo, ON, Canada
[7]Michigan Tech Research Institute, Ann Arbor, MI, 48105 USA
[8]University of Guelph, Guelph, ON, Canada
[9]Universidad del Rosario, Bogotá, Cundinamarca, Colombia
[10]NASA Goddard Space Flight Center, Greenbelt, MD, 20771 USA
[11]UAF: Institute of Arctic Biology, University of Alaska Fairbanks, Fairbanks, AK, 99775 USA
[12]Michigan Tech University, Houghton, MI, 49931 USA
[13]Department of Earth System Science, University of California, Irvine, CA, 92697 USA
[14]Institute of Arctic and Alpine Research, University of Colorado, Boulder CO, 80309 USA
[15]Natural Resources Canada, Canadian Forest Service, Northern Forestry Centre, Edmonton, AB, Canada
[16]NASA Langely Research Center, Hampton, VA, 23666 USA.

*Correspondence to*: Stefano Potter (spotter@woodwellclimate.org)



**Abstract.** Fire is the dominant disturbance agent in Alaskan and Canadian boreal ecosystems and releases

large amounts of carbon into the atmosphere. Burned area and carbon emissions have been increasing with

climate change, which have the potential to alter the carbon balance and shift the region from a historic sink

to a source. It is therefore critically important to track the spatiotemporal changes in burned area and fire

carbon emissions over time. Here we developed a new burned area detection algorithm between 2001 -

2019 across Alaska and Canada at 500 meters (m) resolution that utilizes finer-scale 30 m Landsat imagery

to account for land cover unsuitable for burning. This method strictly balances omission and commission

errors at 500 m to derive accurate landscape- and regional-scale burned area estimates. Using this new

burned area product, we developed statistical models to predict burn depth and carbon combustion for the

same period within the NASA Arctic-Boreal Vulnerability Experiment (ABoVE) core and extended

domain. Statistical models were constrained using a database of field observations across the domain and

were related to a variety of response variables including remotely-sensed indicators of fire severity, fire

weather indices, local climate, soils, and topographic indicators. The burn depth and aboveground

combustion models performed best, with poorer performance for belowground combustion. We estimate

2.37 million hectares (Mha) burned annually between 2001-2019 over the ABoVE domain (2.87 Mha

across all of Alaska and Canada), emitting 79.3 +/- 27.96 (+/- 1 standard deviation) Teragrams of carbon

(C) per year, with a mean combustion rate of 3.13 +/- 1.17 kilograms C m$^{-2}$. Mean combustion and burn

depth displayed a general gradient of higher severity in the northwestern portion of the domain to lower

severity in the south and east. We also found larger fire years and later season burning were generally

associated with greater mean combustion. Our estimates are generally consistent with previous efforts to

quantify burned area, fire carbon emissions, and their drivers in regions within boreal North America;

however, we generally estimate higher burned area and carbon emissions due to our use of Landsat imagery,

greater availability of field observations, and improvements in modeling. The burned area and combustion

data sets described here (the ABoVE Fire Emissions Database, or ABoVE-FED) can be used for local to

continental-scale applications of boreal fire science.





## 1 Introduction


Fire is the dominant disturbance agent in boreal forests (Stocks et al., 2003) and places large controls on ecosystem dynamics including vegetation composition and structure, nutrient cycling, permafrost, and carbon cycling (Bonan and Shugart, 1989; Bond-Lamberty et al., 2007; Walker et al., 2019). Fire frequency, intensity and burned area have been increasing in Alaskan and Canadian boreal forests over the last several

decades (Hanes et al., 2018; Kasischke et al., 2010; Veraverbeke et al., 2017), and these trends are expected to continue throughout the 21$^{st}$ century due to a warmer and drier climate (Balshi et al., 2009; Boulanger et al., 2018; Young et al., 2017). Changes to the fire regime have been associated with more severe fires, which burn deeper into the organic soil profile and may be related to large fire years and seasonal timing of burn (Turetsky et al., 2011), although this has not been tested widely. Ultimately, changes in the fire

regime have the potential to transition at least some North American boreal forests from a carbon sink to a source (Dieleman et al., 2020; Li et al., 2017; Walker et al., 2019, Wang et al., 2021). To better understand how changing boreal fire regimes influence carbon dynamics, it is critical to accurately map burned area and estimate resulting carbon emissions over time.

Burned area mapping in Alaska and Canada over long time frames (> 20 years) has primarily been based on digitized maps of fire observations (both by hand and in recent decades using GPS, aerial imagery, and satellite remote sensing) from the Alaska Large Fire Database (ALFD; Kasischke et al., 2002), the Canadian National Fire Database (CNFD; Amiro et al., 2001; Stocks et al., 2003), and more recently the Canadian National Burned Area Composite (NBAC; Hall et al., 2020). These databases are updated annually in

Alaska and Canada, yet substantial uncertainty remains, particularly as the databases go further back in time and aerial and satellite imagery was less prevalent. Of particular importance is the possibility of commission errors because the databases do not typically account for unburned patches of vegetation and water bodies within the fire perimeters, leading to an overestimation of burned area (Skakun et al., 2021). At the same time, the databases are more likely to omit fires due to lost records or missed detections in



earlier decades (Kasischke et al., 2002; Stocks et al., 2003), leading to omissions. Mapping fire perimeters

in recent decades has improved with the use of satellite remote sensing, particularly from 30 meter (m)

Landsat (Epp and Lanoville, 1996) and 500 m Moderate Resolution Imaging Spectroradiometer (MODIS)

imagery. While MODIS imagery is at coarser resolution than Landsat, its multiple acquisitions per day are

highly amenable to burned area mapping, although there are known omission errors due to small (< 100 ha)

burns as well as an overestimation of burned area at the pixel-level due to the relatively coarse 500 m

resolution, which misses some unburned vegetation patches and water bodies (Giglio et al., 2018). Landsat

imagery can largely bypass these issues of spatial resolution (Guindon et al., 2018; Walker et al., 2018),

but the relatively infrequent overpass times and typical cloudy environments in the tundra and boreal biome

result in data gaps, particularly prior to the launch of Landsat 7 (1999) due to data relay issues and limited

tasking.

Traditionally, carbon emissions from wildfires have been calculated as a function of burned area, fuel

consumption and emission factors (French et al., 2011; Seiler and Crutzen, 1980). Carbon emissions in

these models are based on observed relationships between fuel consumption, fire weather, and fuel type.

Current models that are built with this framework include the Wildland Fire Emissions Information System

(WFEIS; French et al., 2011, 2014), the Fire Inventory from NCAR (FINN; Wiedinmyer et al., 2011) and

the Global Fire Emission Database (GFED; van der Werf et al., 2017). In addition to these regional and

global products, there are several model products that provide estimates in boreal ecosystems of Alaska

(French et al., 2002; Kasischke & Hoy, 2012; Tan et al., 2007; Veraverbeke et al., 2015) and Canada (Amiro

et al., 2001; de Groot et al., 2007). Researchers have also made improvements to process-based models'

representation of fire occurrence and effects (Hantson et al., 2016; Rabin et al., 2017; Zhao et al., 2021).

These models can be used to explore causal relationships and have the benefit of estimating how burn rates

and carbon emissions may vary under differing future climate change scenarios.



In addition to simple empirical and process-based models of carbon combustion, several recent studies have implemented statistical techniques to model combustion based on field observations, satellite remote sensing imagery, and other geospatial data (Dieleman et al., 2020; Rogers et al., 2014; Veraverbeke et al., 2015, 2017; Walker et al., 2018). These advances are possible due to the increasing volume of field observations of combustion, and have the advantages of unraveling complex relationships between

combustion observations and geospatial information to extrapolate over space and time. Satellite imagery collected both pre-fire and post-fire has been particularly useful for these techniques, (Hudak et al., 2007; Key and Benson, 2006). Specifically, the differenced normalized burn ratio (dNBR) combines the near and short-wave infrared bands obtained before and after a fire, and the spectral information retained is sensitive to reductions in vegetation and moisture content post-fire. Due to these qualities dNBR correlates relatively

strongly with aboveground biomass loss, but there have been conflicting findings on the strength of the relationship with belowground fire severity, which are particularly important in boreal ecosystems (Kasischke and Hoy, 2012; McGuire et al., 2009). Additional environmental predictors have been combined with dNBR to statistically model above and belowground combustion across Alaska and Canada, including quantified uncertainties (Dieleman et al., 2020; Rogers et al., 2014; Veraverbeke et al., 2015, 2017; Walker

et al., 2018). Veraverbeke (2015) found topographic variables (elevation, slope, northness), pre-fire vegetation cover (% tree cover) and day of burning to be important predictors for both aboveground and belowground combustion, and more specifically the combination of dNBR, day of burning, elevation and tree cover to be the most informative in Alaska. Walker (2018) considered 71 variables associated with topography, permafrost condition, fire severity, fire weather and soil properties, and found that dNBR,

change in pre- and post-fire tree cover, terrain ruggedness, topographic wetness, percent black spruce and percent sand were the most informative for the 2014 Northwest Territories fires. Although these results have been encouraging, extrapolations have been limited to specific regions in Canada and Alaska, and often to specific fire years. It is likely that the inclusion of additional field data across a more representative selection of field locations in Alaska and Canada would improve model fits and allow for extrapolation

over a larger domain and longer time periods.

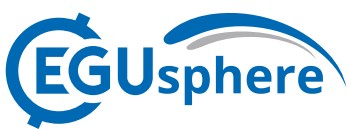

In this study we first derived a new 500 m burned area product for all of Alaska and Canada during 2001-2019. Our approach builds on previous satellite-based burned area mapping efforts (Chen et al., 2020; Dieleman et al., 2020; Loboda et al., 2018; van der Werf et al., 2017; Veraverbeke et al., 2015; Walker et al., 2018) with 500 m MODIS data, but advances these by using 30 m Landsat imagery to both improve accuracy and account for the presence of unburnable land cover. Using this burned area product, along with a new comprehensive database of combustion observations in Alaska and central/western Canada (Walker et al., 2020a), we used machine learning to estimate burn depth and fire carbon emissions across the ABoVE domain. We compare our product to a suite of previous efforts, and use it to test previously hypothesized relationships between fire severity, annual burned area, and seasonal timing of burning.

## 2 Methods

### 2.1 Study Area

The spatial domain of this study includes all of Alaska and Canada for our burned area product, and the ABoVE core and extended domain (hereafter the "ABoVE domain"; Loboda et al., 2019) for our combustion and burn depth product (Figure 1). The combustion and burn depth products were not derived beyond the ABoVE domain due to a lack of field observations in eastern Canada. The temporal domain for all products is 2001-2019. Our study area includes all natural boreal and arctic vegetation within the ABoVE domain, including boreal forests, boreal wetlands, grasslands, tundra, and tundra wetlands. To determine these locations we derived a vegetation mask using the 2005 Land Cover of North America product (250 m; CCRS, 2013; Pouliot and Latifovic, 2013; Pouliot et al., 2014), MODIS land cover type with International Geosphere-Biosphere Programme (IGBP) classification (Collection 6; year 2005; 500 m; Friedl and Sulla-Menashe, 2019), the Circumpolar Arctic Vegetation Map (CAVM; Raynolds et al., 2019) and long-term climate (1970 – 2000; ~1 km; Fick and Hijmans, 2017), all re-gridded to 500 m resolution on the MODIS sinusoidal projection. Boreal vegetation was distinguished from temperate



using a mean annual temperature threshold of 3°C, as recommended in Wolfe (1979) and implemented in

Rogers et al. (2015). Pixels were designated as urban, crop, crop/natural vegetation mosaic, or water if

they were represented as such in either the Land Cover of North America or MODIS land cover products.

Pixels were designated as tundra if they were within the CAVM domain.

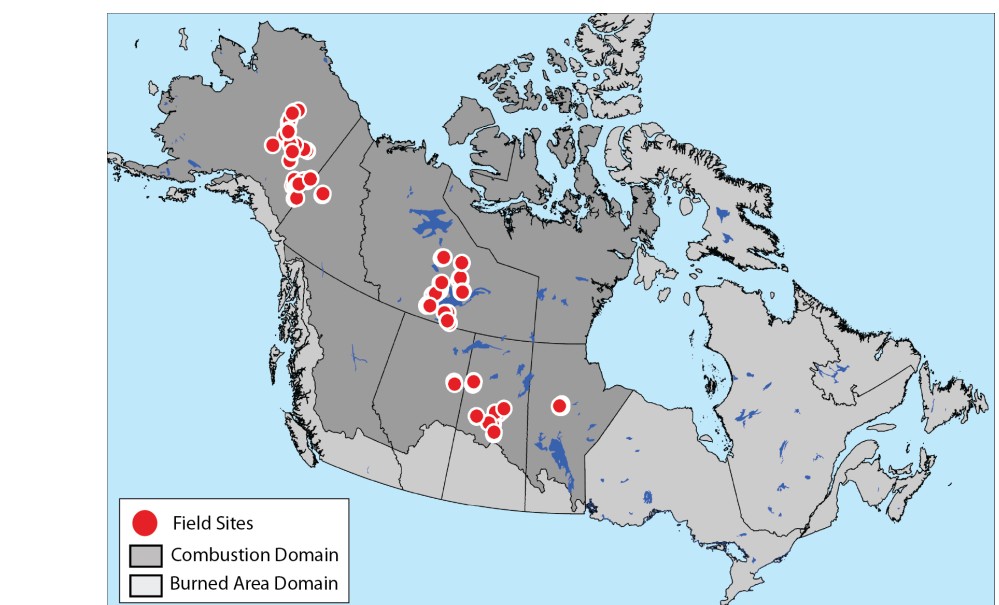

Figure 1. Study domain. Locations of combustion observations are shown in
red, the burned area product domain is shown in light gray, and the
combustion and burned depth product domain is shown in dark gray.


### 2.2 Field Data

Field measurements of burn depth and combustion were derived from numerous data sources across

different research groups that represent a major synthesis effort sponsored by the NASA ABoVE program

(Boby et al., 2010; Dieleman et al., 2020;  de Groot et al., 2009; Hoy et al., 2016; Rogers et al., 2014;

Turetsky et al., 2011; Veraverbeke et al., 2015; Walker et al., 2018). Detailed descriptions of data

collection methods can be found in the contributing publications. All field site information was

standardized and aggregated into a single publicly-available database (Walker et al., 2020a), which has



been used to assess patterns and drivers of ecosystem structure and combustion across ecoregions (Walker et al., 2020b; Walker et al., 2020c). Although the field database only includes measurements from boreal

ecosystems, our combustion and burn depth predictions include both boreal and tundra ecosystems. Of all the pixels for which we predicted combustion and burn depth, only 0.78% are in tundra landscapes.

## 2.3 Burned Area Mapping

The ABoVE Fire Emissions Database (ABoVE-FED) burned area product is derived from a dNBR

thresholding approach, which has previously been successfully employed for burned area mapping in the region (Rogers et al., 2014; Veraverbeke et al., 2015; Walker et al., 2018). Our primary approach was to use Landsat imagery to separate burned from unburned pixels at 30 m. However, because Landsat imagery was not available for all regions and time periods, we used MODIS imagery to map burned pixels when necessary, and upscaled our Landsat-based product to 500 m MODIS resolution. More specifically we used

pre- and post-fire near infrared (NIR) and shortwave infrared (SWIR) bands from Aqua (MYD09GA Collection 6; Vermote, 2015a) Terra (MOD09GA Collection 6; Vermote, 2015b) and Landsat 5-8, calculating dNBR as the difference in pre fire normalized burn ratio (NBR) and post fire NBR, where NBR is near infrared minus shortwave infrared divided by near infrared plus shortwave infrared.

This approach had the added advantage of accuracy; whereas a Landsat dNBR threshold tends to be surpassed at the site level in a diffuse manner across the landscape, due to stochastic site-level disturbances such as tree mortality, herbivory, flooding, or small-scale dieback, it is much less common for these small-scale disturbances to influence the majority of a 500 m pixel. We also minimized mapping non-fire disturbances by following the approach of Veraverbeke et al. (2015) and applying our dNBR approach to

(1) mapped fire polygons from the ALFD and CNFD (93% of total burned pixels; hereafter collectively referred to as the National Large Fire Databases (NLFD)) and (2) MODIS active fire (MOD14A1 Collection 6 and MYD14A1 Collection 6; Giglio et al., 2018) acquisitions outside these polygons (7% of total burned pixels). In each case we applied a 1 km buffer (Veraverbeke et al., 2015) to capture burned pixels



immediately outside these areas. Finally, our approach is motivated by a desire to balance commission and

omission errors at both the 30 m and 500 m scales, thereby providing an unbiased estimate of total burned

area.

To map 30 m burned pixels, we first extracted dNBR at all burned and paired control sites in our aggregated

field database using available cloud-free Landsat 5, 7 and 8 Tier 1 surface reflectance images in Google

Earth Engine (Gorelick et al., 2017). Landsat 5 and 7 were atmospherically corrected using the Landsat

Ecosystem disturbance adaptive processing system (LEDPAS; Schmidt et al., 2013) while Landsat 8 was

atmospherically corrected using Land Surface Reflectance Code (LaSRC; Vermote et al., 2016). Pre- and

post-fire normalized burn ratio (NBR) was calculated as the mean of all available Landsat observations

between July and August. Pre-fire values were extracted one year before a given fire, and post-fire values

were extracted one year after a fire. We then selected a 30 m Landsat dNBR threshold that most effectively

separated burned and unburned control sites. Because there are many fewer unburned control sites in the

Walker et al (2020a) combustion database, we derived additional control sites by extracting dNBR at burned

sites two years before a given fire, which had the advantage of controlling for any site-level spectral

differences between burned and control sites represented in the database. This process generated a dNBR

threshold of 0.084, which minimized 30 m site-level commission and omission errors to 6.6% (Figure S1).

We then created a mask at 30 m to account for unburnable land cover (i.e., non-vegetated pixels). This was

created using two sources: the Joint Research Center's yearly water history product (Pekel et al., 2016) and

the North American Land Change Monitoring System's (NALCM) 2010 land cover product at 30 m

resolution (Latifovic et al., 2012). The first product allowed us to capture transient water pixels in our time

series while the NALCM land cover product classified each pixel into 19 different land cover classes, from

which we masked out non-vegetated pixels including ice, water, barren and cropland. These two sources

were combined into separate masks for each year between 2001-2019. Because areas that burned in 2010



were often classified as barren lands in the 2010 NALCM product, we considered barren lands as vegetated

in our mask for the year 2010.

Using the vegetation mask and the dNBR threshold, we created a binary burned/unburned 30 m Landsat

product and upscaled this to the native MODIS 500 m resolution and projection. To determine whether or

not a given 500 m pixel was classified as burned or unburned, we calculated the percentage of 30 m

vegetated pixels that burned within its footprint. If more than 50% of the 30 m vegetated pixels within the

larger 500 m pixel burned (i.e., were tripped by the dNBR threshold), the entire pixel was assigned as

burned, and the burned fraction was calculated as the percent of the burnable land cover (vegetation) in the

500 m pixel. Note we did not use the percent of burned 30 m pixels to determine burn fraction within a

given 500 m pixel, primarily because of limitations imposed by frequently missing Landsat imagery

(detailed below).

We used this approach whenever 500 m pixels contained 100% coverage by Landsat imagery at 30 m.

When, however, there was less than 100% Landsat coverage, we needed to determine if it was more accurate

to classify 500 m pixels using Landsat (with partial coverage) or MODIS Collection 6 imagery (Vermote,

2015a, Vermote, 2015b). To do so, we analyzed all MODIS pixels with complete Landsat coverage and

masked out increasing numbers of Landsat pixel strips within the larger MODIS footprint (using increments

of 5%). After each removal of Landsat pixels, we compared the accuracy of the resulting burned/unburned

classification using (i) Landsat imagery with partial coverage and (ii) MODIS imagery. This procedure

suggested that using MODIS dNBR was more accurate than Landsat when less than 85% of a 500 m

MODIS pixel was covered by Landsat imagery. We therefore used Landsat to classify burned pixels when

at least 85% of a 500 m pixel was covered by Landsat imagery, and otherwise used MODIS. Burned pixels

were assigned a quality flag of zero when there was complete Landsat coverage, a quality flag of one when

Landsat coverage was less than 100% but greater than 85%, and a quality flag of two when Landsat

coverage was less than 85% and therefore MODIS imagery was used to classify burn status. Overall, 81%





of total burned pixels were derived using Landsat (66% from full coverage and 15% from partial coverage),

although particular regions (notably Alaska and Newfoundland and Labrador) tend to rely more on MODIS

due to more limited availability of Landsat imagery (Figure S2).

We developed a correction factor for MODIS-based dNBR to account for differences between Landsat and

MODIS NIR and SWIR spectra, as well as the influence of vegetation fraction on 500 m dNBR signals. To

do so, we calculated pre- and post-fire NIR and SWIR bands from MODIS and Landsat (resampled to 500

m) for a 50% random sample of burned pixels. We then differenced the Landsat 500 m resampled bands

from the 500 m MODIS bands and regressed them onto vegetation fraction to obtain a correction factor.

The regression yielded an $R^2$ of 0.74 and an equation of $y = 0.94x + 0.01$, which was applied to all pixels

where burn status was classified by MODIS. We then calculated a new dNBR threshold to classify pixels

at 500 m in an unbiased manner. To do so, we determined the MODIS dNBR threshold that evenly split

omissions and commissions based on pixels mapped with complete Landsat coverage.  This threshold was

determined to be 0.0725, resulting in an omission/commission error of 14.2% at 500 m when using MODIS.

One issue with a burned area mapping approach such as ours that utilizes post-fire imagery one year after

a fire is that it is difficult to determine the year(s) of burn where overlapping burns occurred in successive

years. To address these cases, we created a seasonal MODIS-based product following the methodology of

Giglio et al. (2018). The dNBR for each day between January 15[th] and December 15[th] was calculated using

the 30 preceding days as pre-fire NBR and the 30 days after as post-fire NBR. Any pixels with less than 10

valid observations in either window were masked out. We used a similar thresholding approach to that

described above for mapping burned pixels with MODIS, resulting in a seasonal dNBR threshold of 0.23.

Any pixel mapped using the MODIS seasonal approach was assigned a quality flag of three.

In addition to determining fire locations, fire year, and the burned fraction, we also determined the day of

burning for each pixel. When possible, day of burn was taken directly from the thermal anomaly active fire



detections from MOD14A1 Collection 6 and MYD14A1 Collection 6 (Giglio et al., 2018) active fire

products. Where an active fire was registered, day of burn was assigned by taking the earliest active fire

acquisition during the year. When an active fire was not registered for a given burned pixel, we utilized a

multi-tiered approach to assign day of burn. When possible, we used a kriging technique to interpolate day

of burn using the active fire detections within each fire polygon in the NLFD following Veraverbeke et al.

(2015). To implement this, we required fire polygons to contain at least five active fire acquisitions within

their boundaries and have some level of temporal variation (i.e., not all active fire acquisitions on the same

day). When this was not the case, day of burn was assigned using the closest active fire pixel. Finally, when

no active fire acquisitions were associated with a given fire polygon, we used our MODIS-based seasonal

mapping approach to determine day of burn by locating the day of maximal dNBR within a given year. For

fires that were detected by MODIS thermal anomalies but were not contained in the NLFD (7% of all

burned area), we created our own polygons around the burned pixels (by converting pixels to vectors and

buffering them) and used the same method to assign day of burn. Quality flags for our burn day product

represent this tiered approach, with a flag of zero for pixels with direct active fire hits, a flag of one for

pixels whose day of burn was determined by interpolation, and a flag of two for pixels whose day of burn

was determined using the MODIS seasonal burned area product.

We compared ABoVE-FED burned area to several other products including the NLFD, NBAC,

MCD64A1 Collection 5, MCD64A1 Collection 6, the Alaska Fire Emissions Database version 2

(AKFED; Veraverbeke et al., 2017), GFED4s (van der Werf et al., 2017) and the Fire Model

Intercomparison Project (FireMIP; Hantson et al., 2016; Rabin et al., 2017; Table S1). NBAC is a

Canada-only product and is related to the CNFD, but improves upon it by incorporating multi-sensor

remote sensing imagery to account for water bodies and unburned vegetation patches. FireMIP includes

simulations performed by coupled fire-vegetation models forced with a standardized set of input data. We

also visually compared our product and others to high-resolution imagery of fires from the Worldview 2



(1.84 m) satellite, available through DigitalGlobe, Inc., a Maxar Company under the Nextview license
agreement through the National Geospatial Intelligence Agency (Neigh et al., 2013).

### 2.4 Combustion and Burn Depth Models

We built and applied statistical models of aboveground combustion, belowground combustion, and burn
depth to every mapped burned pixel in the ABoVE domain based on field observations across Alaska and
western Canada (Walker et al., 2020a). Because not all field sites included estimates of both above- and
belowground combustion, we created two separate combustion models, one utilizing all available
aboveground combustion measurements (n = 515) and one utilizing all available belowground

combustion measurements (n = 769). Our burn depth model utilized the same field sites as belowground
combustion. Further discussion of models implemented can be found in the Supplement.

### 2.4.1 Predictor variables

Combustion and burn depth measurements from Walker et al. (2020a) were related to a variety of spatial

predictors including remotely sensed indicators of fire severity, topography, soils, climate and fire
weather. We initially acquired 75 covariates associated with environmental conditions such as long-term
climate, fire weather, topography, vegetation type, soil type, remotely sensed vegetation indices (e.g.
normalized difference vegetation index; NDVI; Tucker, 1979) and permafrost condition (Table S2).

### 340 2.4.2 Climate Variables

Long-term climate was acquired from ClimateNA (CNA; Wang et al., 2016; Table S2), which provides
point estimates of mean climate from 1981 – 2010 based on the Climate Research Unit (CRU; Mitchell
and Jones, 2005). ClimateNA uses finer-resolution PRISM (Daly et al., 2002, 2008) and ANUSPLIN
(Hutchinson, 1989) climate normals to downscale coarse-resolution monthly climate data to a 4 × 4 km

grid, followed by bilinear interpolation and a locally-derived elevation adjustment to estimate point data.
CNA variables were represented as both annual and summer means (June – August), and were included to



capture the influence of long-term climate on vegetation, fuel loads, and fuel moisture that drive

combustion (Walker, et al., 2020b).

**2.4.3 Fire Weather Indices**

Fire weather indices (FWIs) represent the meteorology at the timing of fire occurrence and have been

associated with fire behavior and carbon emissions due to their influence on fuel moisture and fire spread

(e.g. Di Giuseppe et al., 2018; French et al., 2011; Ivanova et al., 2011; Veraverbeke et al., 2017). We

acquired FWIs from the Global Fire Weather Emissions Database (GFWED v2.0; Field et al., 2015) at

$0.5° \times 0.66°$ resolution. FWI information was extracted for the day of burn for all fires in the field

database. Since FWI data were not available for all burned pixels in our fire product due to missing data

in the shoulder seasons, we developed two versions of our aboveground combustion, belowground

combustion, and burn depth models: a primary model that included FWIs in training and a secondary one

that did not. Mapped pixels from the primary model were assigned a quality flag of zero, and pixels from

the secondary model were assigned a flag of one. Of the 2,123,730 pixels that burned between 2001 –

2019, 4.4% did not have FWI data available and necessitated the use of these secondary models.

**2.4.4 Environmental Variables**

We acquired a variety of environmental covariates related to soils, topography, vegetation type, and

permafrost occurrence (Table S2). Soil properties were taken from SoilGrids at 250 m resolution (Hengl

et al., 2017), including percent clay (0 – 2 μm), silt (2 – 50 μm), sand (50 – 2000 μm), coarse material ( >

2000 μm), bulk density (g cm$^{-3}$), soil organic carbon stock (tons ha$^{-1}$), and soil water pH. We integrated

all variables across the top 30 cm of the soil profile.

Topographic variables, including elevation (m), aspect (°), and slope (°), were derived from a 10-meter

digital elevation model (DEM) of the ABoVE domain which, in turn, was derived from a higher

resolution Arctic DEM (Porter et al. 2018) and gap-filled with additional DEM datasets (Burns et al.,



forthcoming). This 10 m DEM was resampled to 500 m and then aspect and slope were both calculated

as the local gradient of the four connected neighbors of each pixel. After resampling to 500 m we also

calculated a topographic wetness index (TWI) for each pixel that represents soil drainage patterns based

on the slope and upslope area draining through a particular point (Beven & Kirkby, 1979).

Vegetation type was represented by the percent cover over seven broad classes including black spruce

(*Picea mariana*), white spruce (*Picea glauca*), jack pine (*Pinus banksiana*), deciduous broadleaf species,

other conifers, grasslands and non-vegetated (Beaudoin et al., 2014; Ottmar et al., 2007). We use pre-fire

tree cover (Sexton et al., 2013) from either 2000, 2005, 2010, or 2015, depending on fire year.

 Lastly, we acquired a permafrost zonation and a surface roughness index, which is a measure of terrain

complexity (Gruber, 2012).

**2.4.5 Remotely Sensed Variables**

We derived numerous remotely-sensed vegetation indices from Landsat, including the NDVI, the

normalized difference infrared index (NDII; Hardinsky & Smart, 1983), dNBR (Key and Benson, 2006),

the relative difference normalized burn ratio (RdNBR; Miller and Thode, 2007), the relativized burn ratio

(RBR; Parks et al., 2014), dNBR, tasseled cap greenness, wetness and brightness (Kauth and Thomas,

1976), and pre-fire tree cover (Sexton et al., 2013). NDVI, NDII, and tasseled cap indices were acquired

as a mean composite between May 15th and June 15th in the post-fire years, while dNBR, RdNBR and

RBR were based on mean composites between June 1st and August 31st for both the pre- and post-fire

years.

For model training all remotely sensed variables were extracted from Landsat 5-8 Tier 1 surface

reflectance at 30 m with clouds, cloud shadows and snow masked out using the C Function of Mask

(CFMask; Foga et al., 2017). We applied corrections due to spectral differences between Landsat 8 and 7

using a regression technique (Roy et al., 2016). Although our model was trained with Landsat imagery at





30 m, we predicted combustion and burn depth at 500 m across the domain using MODIS imagery. All

MODIS variables were extracted in Google Earth Engine at ideal MODIS quality flags (bit flag of 0). We

then implemented a correction factor to account for sensor and spatial scaling issues in model predictions

(section 2.4.7).

### 2.4.6 Feature Selection and Model Comparisons

We reduced our initial 75 covariates to an optimal number using recursive feature elimination (Guyon et

al., 2002). Recursive feature elimination iteratively removes variables until a desired number remains,

which in this case is defined by the number of covariates necessary to achieve the minimum root mean

square error (RMSE). Recursive feature elimination achieves this by fitting a secondary machine learning

model that can rank features by importance and discards the least important ones at each iteration. We

used random forest (Breiman, 2001) as the measure of importance, and repeated our recursive feature

elimination three times across a five-fold cross-validation to determine the optimal subset of covariates

(Table S2).  For the primary aboveground combustion, belowground combustion, and burn depth models,

the optimal number of variables was 15, 45, and 40 (Figure S3), respectively, and for the secondary

models the optimal number of variables was 15, 64, and 48. While it is possible a similar RMSE could

have been achieved with reduced model complexity (reduced number of variables), we chose to directly

use RMSE reduction as our threshold for feature selection.

We then tested a suite of statistical models across the selected feature space to compare predictive power.

For each model, we searched for optimal model parameters using a 10-fold cross-validation repeated

three times and a random search grid of length 10 (e.g. for any given model parameter, 10 random

numbers were selected per parameter and tested for each parameter combination). After optimizing model

parameters, we compared final model fits with a 10-fold cross-validation repeated 100 times. After

comparing the median $R^2$ for each model across these 1,000 iterations, we selected the best performing

model and chose it for the final model implementation. All model training took place in R (R Core



Development Team, 2021). In all cases the best performing model was a ranger random forest, although

there were differences in the optimal parameters chosen (Table S3).

### 2.4.7 Spatial Scaling

Our combustion and burn depth models were developed using site-level data (most plots utilized a 30 ×

30 m design) and geospatial predictors at their native resolution, including a variety of 30 m Landsat

indices. However, our spatial model was applied at 500 m to match the resolution of our burned area

product, ultimately because missing imagery prevented comprehensive burned area mapping at 30 m. To

explore potential issues associated with implementing the model at these different spatial scales, we

randomly sampled two hundred 500 m pixels from each year in 2004, 2006, 2012, 2014 and 2015, for a

total of 1,000 pixels. We then implemented our combustion and burn depth models at both 30 m and 500

m to assess biases and errors introduced by both spatial and sensor differences. When models were

assessed at 30 m, all predictor variables were acquired at their native resolutions (Table S2); when models

were assessed at 500 m, all variables were resampled to 500 m. Any variables described in section 2.4.5

that were derived from Landsat were instead collected at 500 m from MODIS (using MOD09A1

Collection 6 and MYD09A1 Collection 6). We used MODIS provided quality flags to select for pixels

that were corrected at ideal quality and masked out clouds and snow. All other variables were resampled

to 500 m using bilinear interpolation if native resolution was >500 m, and using mean values within pixel

boundaries if native resolution was <500 m. We then compared the predictions at 500 m resolution to the

mean across all the 30 m sub pixels, and built type 2 linear regression models to correct for potential

biases. The coefficients from these models were then used to adjust the final predictions for the

combustion models across the full domain.

### 2.4.8 Combustion and Burned Depth Predictions and Quality Flags

Predictor variables for all burn pixels across the domain were collected in Google Earth Engine. Since the

ideal MODIS quality flag criteria (section 2.4.5) left 0.31% of the total burned pixels missing, we




collected predictors for these pixels with no MODIS quality flag applied, and assigned our own quality

flag to distinguish these samples. We provide four separate quality flags indicating whether our primary

or secondary models (no FWIs) were implemented, and whether MODIS quality flags were applied. Our

four flags have the following associations: Flag one – primary model with MODIS quality flag criteria

(95.32 % of pixels); Flag two - primary model with no MODIS quality flag criteria (0.26 % of pixels);

Flag three – secondary model with MODIS quality flag criteria (4.37 % of pixels); Flag four - secondary

model with no MODIS quality flag criteria (0.05% of pixels).

**2.4.9 Monte Carlo Analysis**

To derive a measure of prediction uncertainty, we implemented a Monte Carlo analysis with 500

simulations that incorporated uncertainty from both the field-measured combustion and the random forest

models. Our approach was based on techniques implemented in Rogers et al. (2014), Veraverbeke et al.

(2015), Walker et al. (2018), and Dieleman et al. (2020). To account for uncertainty in field estimates of

belowground combustion, we used the standard error of observed site-level combustion when it was

available. In total, 271 field sites recorded standard error: 22 in Alaska, 47 in Saskatchewan, and 202 in

the Northwest Territories. Standard error was estimated for both aboveground and belowground

combustion in Alaska and Saskatchewan, and only for belowground in the Northwest Territories. For

each Monte Carlo simulation, we derived an adjustment factor by multiplying a site's standard error by a

random number from a normal distribution with a standard deviation of one and centered around zero.

This resulting number was then added to the measured combustion.

Uncertainty in aboveground combustion in the Northwest Territories was calculated by first creating a

random bias for the percent carbon content of trees (central estimate of 0.5), which varied randomly

within a normal distribution with 3% standard deviation systematically across all trees measured for each

Monte Carlo simulation. We similarly included a 20% error in visual estimates of tree consumption



(Dieleman et al., 2020; French, 2004; Walker et al., 2018), which also varied systematically across all trees measured. Aboveground combustion in each simulation was then altered using these adjustment terms (adding the carbon fraction adjuster and multiplying the tree consumption adjuster).


Since these procedures only accounted for uncertainty of 271 of the possible samples, uncertainty for the remaining 245 aboveground and 499 belowground samples was derived using an alternate approach. To do so, we first linearly regressed the aboveground and belowground combustion standard error derived from Monte Carlo simulations against measured aboveground and belowground combustion, respectively.

The coefficients from these two separate models were then used to predict the standard errors for all remaining samples (Figure S4).

In addition to uncertainty in field measurements, there is also uncertainty in the random forest model used to predict combustion across the ABoVE domain. To account for this, we leveraged the fact that model

residual errors tended to increase in proportion to combustion level, similar to Rogers et al. (2014) and Dieleman et al. (2020). To estimate this relationship, we split the original model predictions (from the 10-fold cross-validation repeated 100 times) into 15 bins based on quantiles of total combustion, and then calculated the standard deviation of the residual error within each bin. We then used a general additive model to smooth the standard deviation of the residuals across the bins (Figure S5). For each of the 500

Monte Carlo simulations using adjusted field estimates of combustion (derived from procedures described above), new random forest model predictions were assigned a standard error based on total combustion using the smoothed relationship. These standard errors were then multiplied by a random bias factor with a standard deviation of one centered around zero, which was then added back onto the combustion predictions to derive a final uncertainty estimate for each predicted combustion pixel across the ABoVE

domain.



We quantified uncertainty in our predictions three ways: 1) pixel-level uncertainty, 2) uncertainty in mean combustion and 3) uncertainty in total emissions for a given region of interest. In each case, uncertainties

derived from the Monte Carlo simulations were adjusted by the ratios of mean combustion from the primary model to that of the Monte Carlo simulations in order to account for different mean combustion levels, and hence emissions, between the models (which were minor). 1) Pixel-level uncertainty was calculated as the standard error of combustion for a given pixel across the Monte Carlo simulations. 2) Uncertainty in mean combustion for a given region was calculated as the standard error of mean

combustion across the 500 Monte Carlo simulations for that region. In this case note that mean combustion was calculated by weighting pixels by their vegetated (burned) fractions. 3) Uncertainty in total emissions for a given a region of interest was calculated as the standard error of total emissions for that region across the 500 Monte Carlo simulations.


### 2.5 Relationships between belowground fire severity, annual burned area, and timing of burn

Turetsky et al. (2011) discovered a positive relationship between burn depth, annual burned area, and timing of burn (day of year) in black spruce forests and peatlands of interior Alaska, and also noted the influence of burn timing was more important in small fire years. To test if these relationships held true

with a larger field database in Alaska (n = 286 for ABoVE-FED compared to n = 178 in Turetsky et al. (2011)), we performed a multiple regression of burn depth and belowground combustion using annual burned area and day of year as predictor variables. We also tested how burn depth and belowground combustion varied as a function of day of year within both small and large fire years. To do so, we split the field sites in Alaska into four quantiles based on annual burned area and then regressed burn depth and

belowground combustion against day of year within each quantile. We also conducted this analysis using a sample of 500 ABoVE-FED pixels in Alaska instead of field observations, and then repeated both of these analyses using all available field observations and 500 random pixels within the broader ABoVE





domain. In each case, we sampled 500 pixels instead of using all available pixels to minimize the effect of

large sample sizes on p-values.


## 3 Results

### 3.1 Burned Area

Temporally there was high variability in burned area year to year (Figure 2a). Across the domain,

ABoVE-FED reported similar burned area totals compared to the NLFD (average of 2.87 Mha y$^{-1}$ for

ABoVE-FED compared to 2.90 Mha y$^{-1}$ for NLFD; Figure 3), although there was variability in this

relationship (NLFD estimated larger annual burned area in 11 years and smaller burned area in eight years

between 2001-2019). This was the net result of two contrasting patterns: ABoVE-FED tended to report

less burned area within mapped polygons, due to unmapped unburned patches and unburnable land cover

(e.g., small water bodies) in the government fire databases, but detected additional burned areas

associated with MODIS active fire acquisitions well outside mapped fire polygons (7% of total burned

area in ABoVE-FED; 6% of total emissions; Figure S6). The state/territory with the most burned area

detected outside the mapped polygons was British Columbia (31% of the 7% total burned area mapped

outside NLFD polygons; Figure S6). Exploratory analysis revealed this was likely as a result of

commission errors due to logging (i.e., logged areas tripping dNBR thresholds in conjunction with small

fires registered by MODIS active fire hits). Across the domain, the mean fire size coincident with NLFD

polygons was much larger (4,954 ha) than the mean fire size outside the polygons (166 ha). Because the

NBAC product accounts for more of these unburned patches within polygons (Hall et al., 2020), it tended

to report lower total burned area compared to ABoVE-FED (Figure S7). ABoVE-FED burned area was

substantially higher than MCD64A1 (Collection 5 and 6; Figure 3) in all years, which is consistent with

known omissions in these global products for boreal North America (Giglio et al., 2018; Randerson et al.,

2012; Figure S8). These large-scale patterns were corroborated by high-resolution imagery of particular

fire events (Figure 4; Figure S9). ABoVE-FED identified more burned pixels than MCD64A1 Collection

6 by being more sensitive to fire-induced spectral changes, but also accounted for unburnable portions of



the landscape (Figure S10). GFED4s burned area was slightly higher than both MODIS products, but

lower than the NLFD and ABoVE-FED (Figure 3, Figure S11; average of 2.93 Mha y$^{-1}$ for ABoVE-FED

during 2001 – 2016 compared to 2.38 Mha y$^{-1}$ for GFED). The MCD64A1 Collection 5, Collection 6 and

GFED4s databases underestimated burned area by 32, 23, and 18% compared to ABoVE-FED,

respectively.


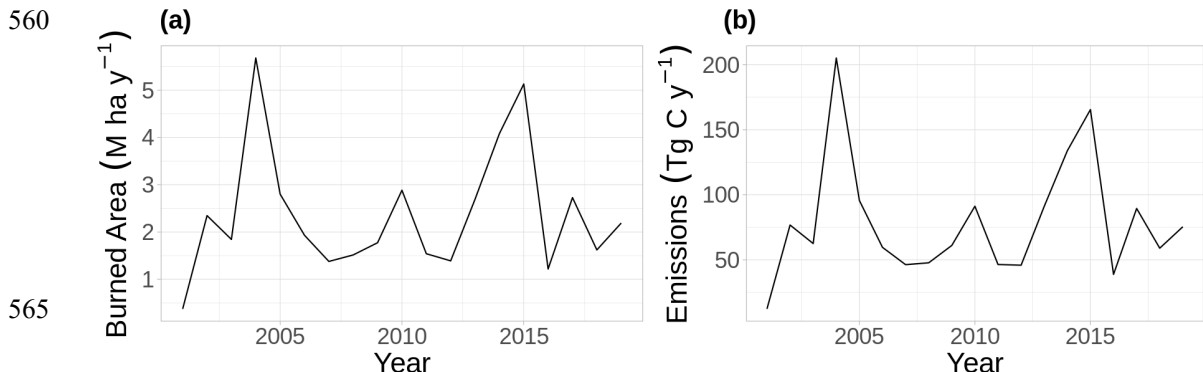


Figure 2. Temporal variability in ABoVE-FED burned area (a), and emissions (b) from 2001-
2019.








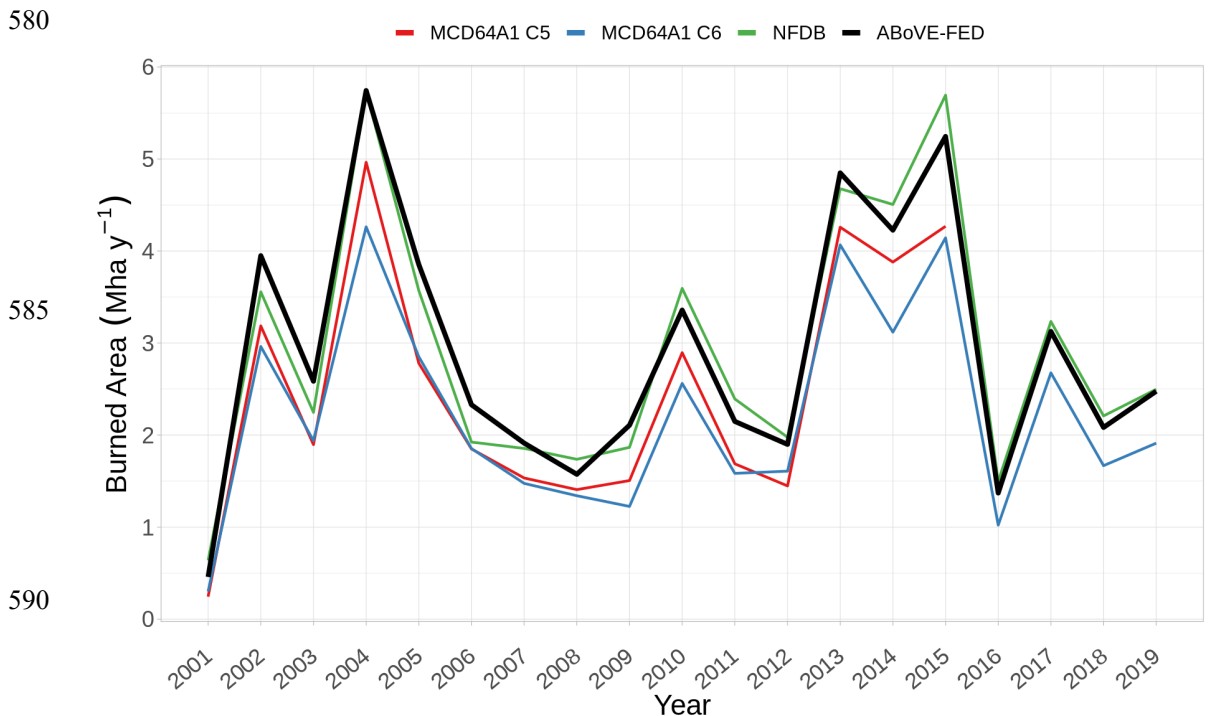

Figure 3. Comparison of ABoVE-FED burned area across Canada and Alaska to MODIS
MCD64A1 Collection 5 (C5), MCD64A1 Collection 6 (C6), and the Alaskan and Canadian
National Fire Databases (NFDB).











Figure 4. Comparison of high-resolution imagery and burned area products for a fire in Manitoba in 2014 (a). Panels show Worldview-2 imagery (b, fire shown in purple shades), ABoVE-FED (c), MODIS Colllection 6 (d), MODIS Collection 5 (e), the Canadian National Fire Database (f) and the National Burned Area Composite (g)



ABoVE-FED burned area was similar to AKFED where it is available (Alaska, the Northwest Territories and the Yukon Territory; Figure S12 ; average of 1.27 Mha $y^{-1}$ for ABoVE-FED during 2001 – 2015 compared to 1.22 Mha $y^{-1}$ for AKFED). All models participating in FireMIP simulated lower burned area than ABoVE-FED, and with a very high level of variability between models ($1.34 \pm 0.83$ Mha $y^{-1}$ across Alaska and Canada during 2001 – 2012; Figure S13a).

Burned area was highly variable interannually, with the largest fire years occurring in 2004 in Alaska and the Yukon Territory; 2015 in Alaska, Saskatchewan, and Alberta; 2014 in the Northwest Territories; and 2013 in Manitoba and Quebec (Figure S11, S14). Across states, provinces, and territories, total burned area was highest in Alaska, the Northwest Territories and Saskatchewan. A total of 54 Mha burned across Alaska and Canada during all years, and 45 Mha in the ABoVE domain, with an annual mean of 2.87 Mha $y^{-1}$ across Alaska and Canada and 2.37 Mha $y^{-1}$ in the ABoVE domain.

Spatially ABoVE-FED estimated the most burned area in in Alaska, the Northwest Territories and Saskatchewan (Figure 5a; Figure S11, Figure S14)



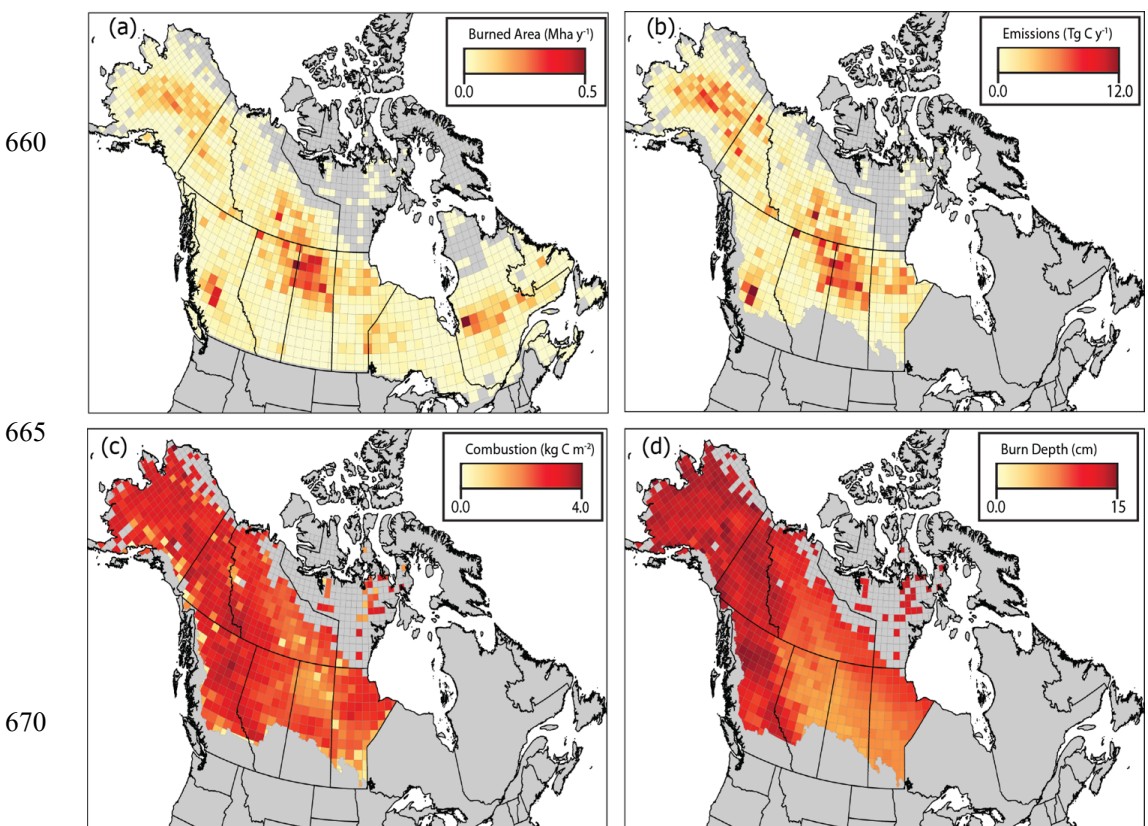

Figure 5. Total burned area (a), total carbon emissions (b), mean combustion (c), and mean burn depth (d) between 2001-2019 aggregated to a 70 km grid. Note that burned area (a) covers all of Alaska and Canada, whereas all other metrics cover the ABoVE extended domain.



### 3.2 Combustion and Burn Depth Models

Our aboveground and belowground combustion models performed well, although the aboveground model

performed significantly better across the suite of models examined (Figure 6a,b). A ranger random forest

model (Wright and Ziegler, 2017) performed best for aboveground and belowground combustion, with

median $R^2$ of 0.46 and 0.25, respectively, across the 10-fold cross validation repeated 100 times. Our

secondary models that did not include information on FWIs (section 2.4.3) performed similarly to our

primary models, with $R^2$ values for above and belowground combustion of 0.45 and 0.24, respectively.

Although both the aboveground and belowground models performed reasonably well at predicting lower

and moderate combustion values, which includes the majority of field observations, they both struggled to

predict larger combustion values (Figure S15a,b). The burn depth model performed better than both

combustion models, with a median $R^2$ of 0.53 using a ranger random forest model (Figure 6c).








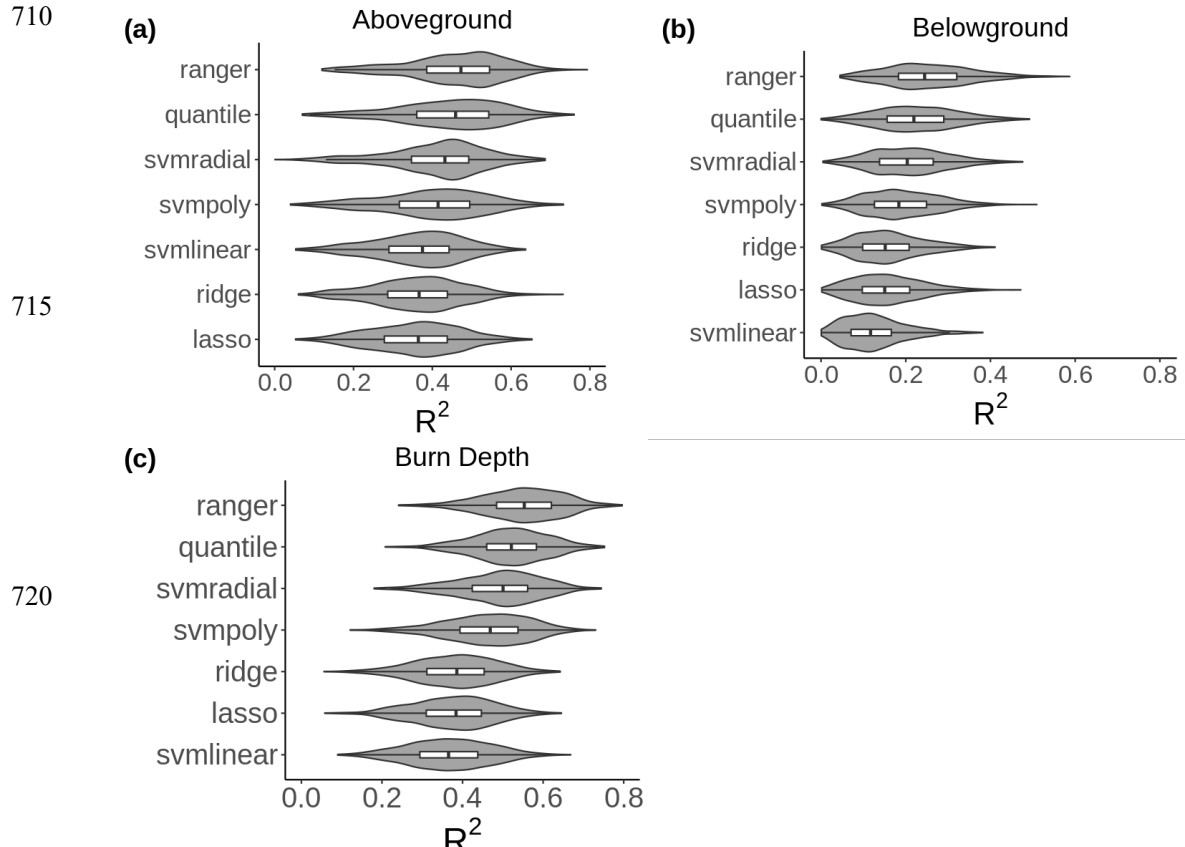

Figure 6. Comparison of the spread and median $R^2$ values across a 10-fold cross validation repeated
100 times for our aboveground combustion (a), belowground combustion (b), and burn depth (c)
models. Models compared include a ranger random forest (ranger), a quantile random forest (quantile
), radial support vector machines (svmradial), polynomial support vector machines (svmpoly), linear
support vector machines (svmlinear), ridge regression (ridge) and lasso regression (lasso).



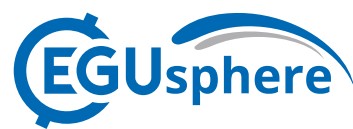

There were notable differences in the feature importance of the aboveground and belowground models (Figure S16a,b). The aboveground model was heavily influenced by its top predictor, pre-fire tree cover, followed by metrics of relative humidity, with other variables including remotely-sensed fire severity and vegetation moisture content having significant but relatively low importance. In contrast, the

belowground model was influenced strongly by a number of soil, terrain, climate, and tree cover variables. The most important features for the burn depth model were similar to the belowground model, with soil properties, tree cover and climate being the most influential (Figure S16c). Overall, the distribution of variables used in the training dataset and predicting dataset were similar (Figure S17), with the exception of slope. Most field sites were located in relatively flat terrain whereas the combustion

predictions included locations with steeper terrain.

Spatial patterns of mean burn depth and combustion tended to follow a gradient of higher burn depth and mean combustion in the western part of the ABoVE domain (Alaska, Yukon Territory, and Alberta) to lower mean combustion in central-western Canada (Saskatchewan, Northwest Territories, and Manitoba)

(Figure 5c,d, Figure S18). There was, however, considerable fine-scale variability at 500 m within these regions (Figure 7), and spatial patterns were relatively consistent with previous combustion mapping efforts.





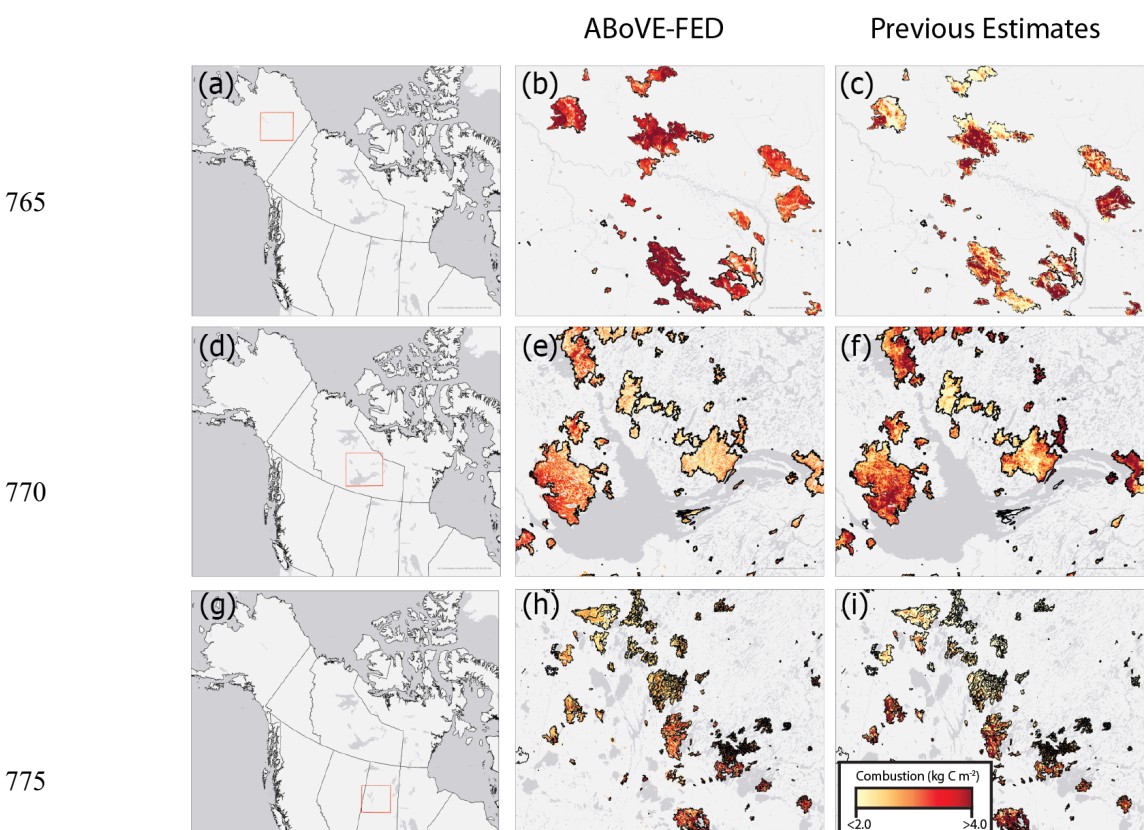

Figure 7. Comparison of Alaskan fires in 2004 (a) for ABoVE-FED (b) and AKFED (c), the Northwest Territories fires in 2014 (d) for ABoVE-FED (e) and Walker et al., 2018 (f), and Saskatchewan fires in 2015 (g) for ABoVE-FED (h) and Dieleman et al., 2020 (i). Basemap Sources: Esri, © OpenStreetMap Contributors, HERE, Garmin, USGS, EPA, NPS, NRCran.

Across the ABoVE domain, 1.51 +/- 0.53 Pg C was emitted over the 2001-2019 period, with a mean of

79.3 +/- 27.96 Tg C per year. Mean combustion across all years and regions was 3.13 +/- 1.17 kg C m$^{-2}$.

Pixel-level uncertainty (Figure S19) tended to follow spatial patterns of mean combustion (Figure 5c) and

was relatively consistent across years (Figure S20), with a mean value of 2.86 kg C m$^{-2}$. Seasonally, the

majority of burned area occurred during June, July and August (Figure 8), although there were substantial

regional differences, with some regions recording a large fraction of burned area outside this window

(e.g., May fires in Alberta). In general, monthly patterns in emissions (Figure S21) followed patterns in

burned area. Overall, combustion tended to be highest in summer compared to spring and fall fires,





although this pattern was most pronounced in the Yukon Territory, Northwest Territories, Saskatchewan

and Alaska (Figure 22).


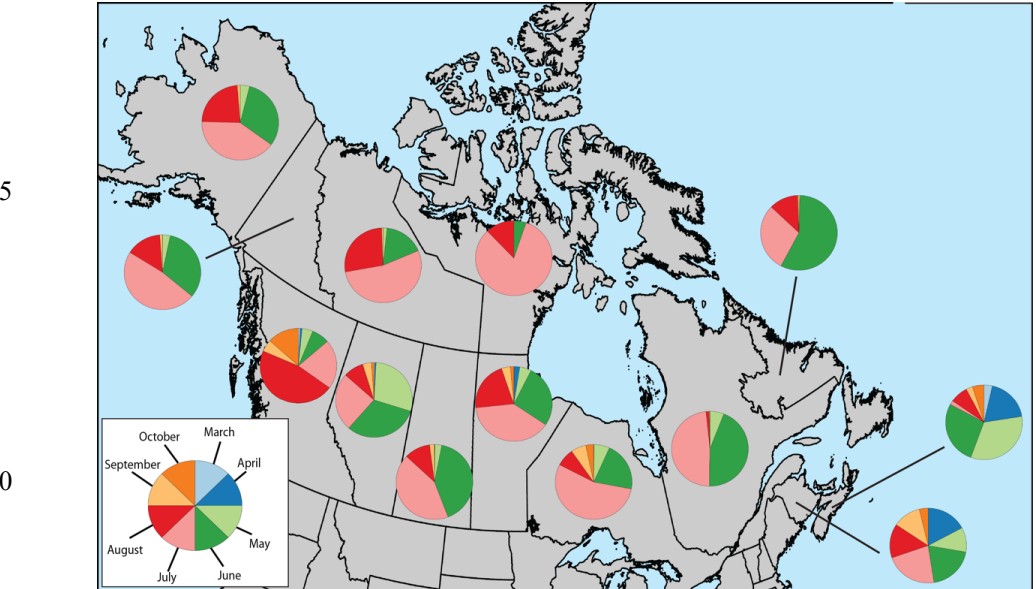

Figure 8. Monthly burned area across states, and Canadian provinces and
territories between 2001-2019. January, February, November and December have
been omitted due to low fire occurrence.





Estimates of total carbon emissions in ABoVE-FED were similar to AKFED (Figure S12; Table S4), with

the notable exception of 2014 in the Northwest Territories: AKFED estimated 164 Tg C and ABoVE-

FED estimated 89.7 Tg C. This was primarily a result of differences in mean modeled combustion in the

Northwest Territories 2014 fires, with AKFED exhibiting its highest mean combustion in 2014 (Figure

S12; 4.81 kg C m$^{-2}$ in AKFED compared to 2.89 kg C m$^{-2}$ in ABoVE-FED). In general, ABoVE-FED

estimated slightly higher mean combustion levels than AKFED in Alaska (3.34 kg C m$^{-2}$ in ABoVE-FED

and 3.03 kg C m$^{-2}$ in AKFED), lower combustion in the Northwest Territories (3.29 kg C m$^{-2}$ in ABoVE-

FED and 3.44 kg C m$^{-2}$ in AKFED), and substantially higher combustion in the Yukon Territory (3.71 kg

C m$^{-2}$ in ABoVE-FED and 2.26 kg C m$^{-2}$ in AKFED) (Figure S12; Table S4). ABoVE-FED carbon

emissions were relatively similar to Walker et al. (2018) for the 2014 Northwest Territories fires, and to

Dieleman et al. (2020) for the 2015 Saskatchewan fires (Figure 7, Table S4). Total carbon emissions from

ABoVE-FED were substantially higher than GFED (Figure S23), with the largest differences occurring in

Alaska. This was primarily a function of higher mean combustion values in ABoVE-FED compared to

GFED (Figure S24). Between 2001-2016, ABoVE-FED estimated 80 Tg C y$^{-1}$ total emissions with a

mean combustion value of 3.39 kg C m$^{-2}$, and GFED estimated 51 Tg C y$^{-1}$ total emissions with a mean

combustion value of 2.30 kg C m$^{-2}$ (Table S4).


Differences in combustion and carbon emissions were very large between ABoVE-FED and fire-

vegetation models participating in FireMIP (Figure S13b). ABoVE-FED estimated much higher

emissions than FireMIP (70.1 Tg C y$^{-1}$ for ABoVE-FED during 2001-2012 compared to 4.0 Tg C y$^{-1}$ for

FireMIP). This is likely because models in FireMIP mostly combust aboveground vegetation, whereas

combustion from belowground sources (primarily soil organic matter) comprises 90% of total carbon

emissions in ABoVE-FED (Figure S25) and 88% in the field plots from Walker et al. (2020a). ABoVE-

FED mean aboveground combustion (7.84 Tg C y$^{-1}$ during 2001-2012) was much more similar to

FireMIP's 4.0 Tg C y$^{-1}$.



We found multiple lines of evidence that belowground fire severity (burn depth and belowground combustion) is positively related to annual burned area and seasonal day of burn (Table S5; S6). In general, mean annual burned area had a stronger relationship with fire severity than did burn day of year using multiple linear regression. However, within quantiles of annual burned area (i.e., low vs. high fire years), day of year was strongly related to fire severity (particularly belowground combustion), and the

slope of this relationship was generally larger in small fire years (Table S6). When assessed using domain-wide mean severity from mapped ABoVE-FED pixels, we found no significant relationship of burned depth with burned area, but combustion increased as a function of burned area (p-value <= 0.10; Figure S26).

There were no significant (p-value <= 0.10) trends in burned area, combustion, or emissions across the 2001 - 2019 time series (Figure 2(a,b), Figure S27).

## 4 Discussion

### 4.1 Burned Area

Our approach to mapping burned area across boreal North America has several advantages compared to past approaches. Although our burned area product is at 500 m resolution, the majority of pixels (81%) were mapped using 30 m Landsat imagery. Using finer-scale 30 m imagery allowed us to directly calibrate dNBR thresholds to site-level information and account for unburnable fractions of 500 m pixels. We also calibrated these dNBR thresholds for both 30 m Landsat and 500 m MODIS imagery to most

effectively balance omissions and commissions. This allowed us to provide an unbiased estimate of burned area, which is a critical variable for understanding the impacts of fire on arctic-boreal ecosystems and climate.

In theory, ABoVE-FED burned area would be expected to be higher than other available products because

of its increased sensitivity to fire-induced spectral changes (compared to, for example, global MODIS





burned area products, via our focus on splitting omissions and commissions) and our accounting for

active fire acquisitions outside mapped fire polygons by the Alaskan and Canadian government agencies.

Alternatively, ABoVE-FED accounts for sub-pixel heterogeneity of burnable land surfaces, which would

otherwise result in lower burned area estimates compared to existing products. The net result is that

ABoVE-FED burned area tends to be higher than other products, but not exclusively.

We suggest future research efforts focused on burned area mapping in arctic-boreal environments could

be conducted at resolutions finer than 500 m. Doing so will allow for improved understanding of fire

spread and behavior patterns, and interactions between fire behavior and vegetation / land cover type.

Finer-scale mapping should also allow for more accurate assessments of burned area by accounting for

the presence of unburned patches of vegetation and water bodies, thereby facilitating increased

understanding of the drivers of fire spread and effects on ecosystem processes (Hall et al., 2020). Fires

have typically been mapped at landscape scales using 500 m MODIS imagery because of the frequent

revisit times (multiple acquisitions per day). With a resolution of 30 m, Landsat imagery has been less

commonly used for mapping burned area at landscape scales because the revisit time (16 days) is much

longer, and because data coverage can be highly variable regionally and spatially depending on available

downlink stations and cloud cover (Hilker et al., 2009; Ju and Masek, 2016; Figure S2), but this revisit

frequency is improving with two Landsat satellites (Landsat 8 and 9) and two Sentinel satellites (2a and

2b) in orbit, which provide much more frequent overpasses (2-3 days when combined).


Similar to ABoVE-FED, approaches for mapping burned area using satellite imagery have typically relied

on image differencing of vegetation indices, particularly dNBR (French et al., 2015). This requires pre-

and post-fire image pairs, and thus compounds issues related to image availability at fine scales (30 m;

(Chen et al., 2021).  Future burned area mapping at landscape scales could potentially be improved by

using machine learning.  More specifically, deep learning approaches have been shown to be highly

effective at mapping wildfires across different landscapes and vegetation types (Jain et al., 2020, Knopp



et al., 2020). Convolutional Neural Networks, which use a spatial moving-window and therefore account

for the spatial characteristics of fire scars (Jain et al., 2020) are particularly promising. Finally,

developing burned area products in near real-time, as opposed to active fire-based assessments of hot

pixel counts, would help scientists, fire managers, and society contextualize and potentially mitigate

rapidly progressing fire seasons as they evolve.

### 4.2 Combustion and Burn Depth Models

Similar to previous studies (e.g., Veraverbeke et al., 2015), our aboveground combustion model

performed substantially better than our belowground model. This is due primarily to the challenge of

estimating belowground carbon consumption using remote sensing-based observations, which are more

sensitive to aboveground properties. For example, the ABoVE-FED aboveground combustion model was

heavily influenced by remotely-sensed properties such as pre-fire tree cover, fire severity (represented by

dNBR), and vegetation wetness (represented by NDII), whereas the belowground model was strongly

influenced by soil metrics, topography and solar radiation (Figure S16).  This occurred despite our model

utilizing considerably more field observations (n = 515 for aboveground combustion and 769 for

belowground) than past efforts in boreal North America (e.g., Dieleman et al., 2020 (n = 47);

Veraverbeke et al., 2015 (n = 126); Walker et al., 2018 (n = 211)), suggesting an inherently limited

capacity to model belowground combustion using these techniques. Previous analysis of the field

observations we used showed site-level drainage is the dominant driver of combustion in the ABoVE

domain, due in part to the large contribution towards total combustion from belowground carbon stocks

(Walker et al., 2018; 2020b). We therefore suggest prioritizing the use of geospatial products that

adequately capture drainage, and thereby its impact on belowground carbon stocks and vulnerability to

combustion, for improving future estimates of carbon emissions from fire disturbance across boreal North

America.



Despite these limitations, our model performance is similar to past efforts. For example, Veraverbeke et al., 2015 reported an aboveground combustion model fit of $R^2 = 0.53$ and a belowground fit of $R^2 = 0.29$ for Alaska. Walker et al. (2018) implemented a 10-fold cross validation approach and reported a model fit

of $R^2 = 0.26$ for total (above and belowground) combustion in the Northwest Territories, Canada. Comparatively, we report a median $R^2$ of 0.46 and 0.25 for ABoVE-FED aboveground and belowground combustion models, respectively. However, model performance was substantially higher in Dieleman et al. (2020), who reported a cross-validated $R^2$ of 0.73 for total combustion in Saskatchewan. This is likely due to the higher relative contribution from aboveground combustion in the younger and more productive

boreal forests of southern Canada, combined with high-quality provincial spatial data sets such as logging history (Dieleman et al., 2020). In all these cases, spatial patterns from ABoVE-FED are generally consistent with previous efforts (Figure 7), lending confidence to assessments of drivers and spatio-temporal patterns of combustion.

Somewhat surprisingly, our models of burn depth performed better than both aboveground and belowground combustion models (cross-validated $R^2 = 0.53$), which is considerably better than the $R^2$ model fit of 0.40 reported for the burn depth model in Veraverbeke (2015). This suggests substantial uncertainty in translating burn depth to carbon emissions in these boreal forests, which underscores the need for improved spatial layers of soil properties such as bulk density (Houle at al., 2017) and carbon

fraction. The field and laboratory techniques used to calculate carbon emissions from burn depth also contain uncertainty, which is not always quantified. These errors are likely compounded when aggregating data across field campaigns, ecozones, and research groups, such as we did here. Nevertheless, burn depth is a critical fire severity property in its own right, with applications ranging from understanding the changing boreal carbon cycle (Walker et al., 2019) to post-fire succession and

vegetation patterns (Baltzer et al., 2021; Johnstone et al., 2010). Our results suggest geospatial statistical modeling is well-suited for capturing and extrapolating depth of burn in organic soils, at least within the ABoVE domain.



Finally, we assessed the influence of spatial and sensor differences when building the combustion and

burn depth models at 30 m but predicting them at 500 m. Overall, biases introduced by model

nonlinearities, sub-grid heterogeneity, and vegetation fractions were found to be negligible (slope = 0.98

for aboveground and 0.97 for belowground combustion when regressing 500 m against aggregated 30 m

predictions). This suggests that approaches to map fire carbon emissions at large scales using 500 m

MODIS imagery are not fundamentally biased because of spatial scale.


The machine learning models we employed allow insights into the drivers of both aboveground and

belowground combustion. Partial dependence plots indicated that aboveground combustion tended to

increase when tree cover and dNBR increased, and when relative humidity and vegetation water content

(NDII) decreased (Figure S28). These patterns are consistent with understanding of fire behavior and

aboveground consumption dynamics, which are generally driven by aboveground fuels and climate

conditions that facilitate fuel drying and fire spread (Beck et al., 2011; Rogers et al., 2014; Walker,

2020b). Alternatively, belowground combustion increased with higher silt (and lower sand) content,

higher tree cover, and lower relative humidity (Figure S29). At moderate slopes (less than 20%), in which

the majority of field observations were located, belowground combustion was higher in flatter landscapes.

These relationships are consistent with current understanding about the drivers of soil organic matter

accumulation and vulnerability to combustion (Walker et al., 2018, 2020b; Scholten et al., 2021). Drivers

of burn depth were similar to those for belowground combustion, with the exception of higher burn depth

occurring in areas with lower extreme maximum temperatures and Tasseled Cap Greenness (Figure S30).

The former is likely related to deeper burn depths occurring in the northern portions of the ABoVE

domain (Figure 5d), where long-term maximum temperatures are generally lower. Tasseled Cap

Greenness was assessed after a given fire, and can therefore be considered a metric of fire severity (low

greenness = high severity).





Total emissions from ABoVE-FED are relatively consistent with past efforts, including AKFED and

GFED, but with some important differences. Total emissions and mean combustion (Figure S12) in

Alaska were similar between ABoVE-FED and AKFED, which is expected given the similar field

observations from Alaska used to develop these models. However, although AKFED was extended to the

Yukon and Northwest Territories (Veraverbeke et al., 2017), it did not incorporate field observations from

these regions. By utilizing 797 field plots across these provinces (albeit heavily dominated by the

Northwest Territories), our results suggest AKFED tended to underestimate combustion in the Yukon and

overestimate combustion in the Northwest Territories, especially during the large fire year of 2014.

ABoVE-FED also includes many more predictor variables than AKFED, and is based on a different

statistical model. We did not find large variations in mean combustion from year to year (Figure 2), which

is likely related to both the tendency of the random forest models to regress to the mean (Figure S15) as

well as relatively consistent observed mean combustion across large regions of the ABoVE domain

(Walker et al., 2020a, c).

GFED is a widely-used data source for global and regional burned area and fire emissions. Our results

suggest GFED underestimates combustion across the ABoVE domain by roughly 1/3rd (32%; Figure

S24; mean of 3.39 kg C m$^{-2}$ in ABoVE-FED compared to 2.30 kg C m$^{-2}$ in GFED), leading to 36% lower

total emissions compared to ABoVE-FED (Figure S23). This is consistent with previous regional studies

noting a consistent underestimation for GFED emissions in Alaska (Veraverbeke et al., 2015) and the

Northwest Territories (Walker et al., 2018). This result has important implications for quantifying and

understanding the role of arctic-boreal fires on the global carbon cycle and climate. Regional- to

continental-scale upscaling efforts such as ABoVE-FED, including the underlying field observation

database (Walker et al., 2020a), can help inform further versions of global fire models and thereby

improve our quantification and understanding of the role of wildfire on the global carbon cycle.



In contrast to AKFED and GFED, fire carbon emissions in FireMIP were an order of magnitude lower

(94%) than ABoVE-FED (Figure S13b). This is likely due to the fact that most models in FireMIP only

combust aboveground vegetation, whereas combustion of belowground soil organic matter constitutes the

majority of emissions in boreal Alaska and Canada. This underscores the importance of developing

algorithms that accumulate and burn soil organic matter within global fire models, which is important for

both direct fire emissions as well as post-fire permafrost thaw and degradation (Genet et al., 2013; Jafarov

et al., 2013; Natali et al., 2021; Treharne et al., 2022).

ABoVE-FED confirms the high interannual variability of fire carbon emissions in the ABoVE domain,

including the large fire years of 2004 in Alaska and the Yukon Territory, 2005 in Alaska, 2010 in

Saskatchewan, 2014 in the Northwest Territories, and 2015 in Alaska and Saskatchewan. We also found

general agreement with previous work (Turetsky et al., 2011) that large fire years and later seasonal fires

facilitate deeper burning and higher belowground carbon emissions, including the phenomenon that burn

timing has a stronger influence on severity in low fire years (i.e., extreme fire years result in high severity

regardless of timing). However, these relationships varied depending on region and analysis technique,

and were often confounded by site-level factors and fire weather at the time of burn. Overall, however,

this underscores the influence that climate change (warming, drying, and longer fire seasons) has on

boreal fire severity.

Consistent with previous studies (Rogers et al., 2014; Veraverbeke et al., 2015; Walker et al., 2018;

Dieleman et al., 2020), ABoVE-FED includes high uncertainty in combustion at the pixel-level (2.86 kg

C m$^{-2}$). Much of this uncertainty likely arises from difficulty in predicting large combustion values,

particularly from belowground sources (Figure S15b). This suggests ABoVE-FED is underpredicting

emissions coming from the most severe fire events between 2001-2019. We attempted to correct for this

bias in a number of ways, including testing a variety of models (Figure 6), tuning model parameters,

assigning higher weights to the highest combustion values, and applying the Synthetic Minority





Oversampling Technique (SMOTE; Chawla et al., 2002) to synthetically create more samples with higher

combustion values. Ultimately, none of these approaches were able to correct for the low bias at high

combustion levels without sacrificing performance for low combustion values. More field observations of

high combustion combined with improved predictor variables (particularly drainage) may improve future

model performance. Also consistent with previous studies, these pixel-level uncertainties were dampened

through spatial averaging, such that domain-wide mean combustion had comparatively lower uncertainty

(3.13 +/- 1.17 kg C m$^{-2}$).

**Conclusions**

Here we used 30 m Landsat and 500 m MODIS imagery to map burned area across Alaska and Canada,

and map fire carbon emissions across the ABoVE domain over a 19-year period between 2001-2019.  We

utilized a recent field database of combustion observations across the ABoVE domain (Walker et al.,

2020a), which represents the largest of its kind for any biome on Earth. We found burned area and total

emissions are highly variable by year, averaging 2.37 Mha of burned area and 79.26 +/- 28.65 Tg C

emitted per year across the ABoVE domain (2.87 Mha of burned area across all of Alaska and Canada),

with a mean combustion level of 3.13 +/- 1.20 kg C m$^{-2}$. When compared to previous products we report

more burned area than GFED and the MODIS MC64A1 Collection 5 and 6 products. We report similar

carbon emissions to AKFED, but more emissions than both GFED and FireMIP. ABoVE-FED can be

used to understand patterns of fire behavior and effects across central and western boreal North America,

and to continue monitoring intensifying fire regimes in boreal forests.


**Code and Data Availability**

The burned area, combustion and burned depth databases associated with this publication can be
found at Potter et al. 2021 (https://doi.org/10.3334/ORNLDAAC/2063
). Code is available upon request from the corresponding author.


**Competing Interests**





**Author Contribution**

S.P, and B.M.R. contributed to original draft writing. S.V., X.W., M.C.M., S.J.G., J.B., L.B.C., N.F.,

E.E.H., L.J., J.F.J, E.S.K., S.M.N., J.T.R., M.R.T., B.M.R. contributed to conceptualization. S.P., S.C.,

S.H., and B.M.R. contributed to formal analysis. S.V., X.W., M.C.M., S.J.G., L.B.G., N.F., S.M.N.,

J.T.R., M.R.T., and B.M.R. contributed to funding acquisition. S.P., S.C., S.V., X.W., J.T.R. and B.M.R.

contributed to investigation. S.P., S.C., S.V., X.W., M.C.M., S.J.G., J.T.R. and B.M.R contributed to

methodology. B.M.R. was project administrator. E.E.H. contributed project resources. S.V., X.W.,

M.C.M., S.J.G., S.M.N., J.T.R. and B.M.R. contributed to project supervision. S.P. and S.C. contributed

to data curation, validation and visualization and software development. All authors contributed to the

writing-review process.

**Acknowledgements**

This work was funded by the National Aeronautics and Space Administration (NASA) Arctic-Boreal

Vulnerability Experiment (ABoVE grants: NNX15AU56A and NX15AT71A, B.M.R. and M.C.M.;

NNX15AT83A and 80NSSC19M0107, L.L.B.C, N.H.F.F., L.K.J), the Gordon and Betty Moore

Foundation (grant #8414), the Woodwell Climate Research Center's Fund for Climate Solutions, and the

Department of Defense (DoD) Strategic Environmental Research and Development Program (SERDP

contract RC18-1183)

Computing resources for this work were provided by the NASA High-End Computing Program through

the NASA Center for Climate Simulation at Goddard Space Flight Center. S.V. acknowledges support

from the Dutch Research Council through Vidi grant 016.Vidi.189.070 and from the European Research

Council under the European Union's Horizon 2020 research and innovation programme (grant agreement

No 101000987). In kind support was provided through Bonanza Creek LTER with funding from the

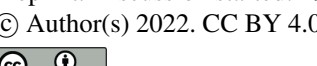



National Science Foundation (DEB-1636476) and the USDA Forest Service, Pacific Northwest Research

Station (RJVA-PNW-01-JV-11261952-231).

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
