# Peer review of "Burned Area and Carbon Emissions Across Northwestern Boreal"

_EGUsphere, 2022_

## Author Comment (AC1)

In this work burnt area maps and carbon emissions are derived for Canada and Alaska for a period of 19 years, based on satellite imagery and a large data base of field observation. Concerning the carbon combustion estimation, both aboveground models and belowground models are used. The latter is very important since in the Canadian and Alaskan boreal ecosystems fire typically burns deeply into the organic soil layer, which limits the application of remotely sensed data. Results area compared with several data sources.

We thank the reviewer for their helpful comments. We have addressed each and believe the manuscript is stronger as a result. Original comments are given below, and our responses are in blue text.

Major comments:

1. Although the manuscript is well structured and clear, due to the large number of analysis performed, it would be easier to fallow the work done if a flowchart was available, describing the methods, data, variables, etc. Or two flowcharts, one for the burnt area mapping model, and one for the combustion and burn depth models.

   We agree with the comment and have added two flow charts, one for burned area (Figure S4) and one for combustion/burn depth (Figure S8), which we believe will help readers follow the methodology.

2. Burned area estimates are compared with other products (regional, from Canada, and global, such as MODIS), and total annual burnt area values are very similar (Fig. 3), but it would be useful to actually validate the burnt area maps. If possible, at least for the areas covered with Landsat, the burnt area maps could be validated with manually digitized burnt area maps over Landsat images, or with Sentinel-2 burnt images for the last years.

3. If the validation suggested in 2 is not possible, a sample of individual fires could be selected and the different burnt area maps discussed (as done in Fig. 4, and Fig. S10), to better understand the impacts (e.g. commission and omission errors) of the different methods.

   We appreciate the reviewer's comment regarding validation. However, we should stress that the National Burned Area Composite product in Canada (NBAC) uses 30 m Landsat as well as finer scale aerial imagery and QuickBird (now MAXAR) (lines 322-325). Hence, comparing our product to NBAC provides a reliable measure of accuracy. Nevertheless, we agree that comparing our product to high-resolution commercial imagery provides valuable comparative insight. To address this, we added seven new figures (Figures S13 – S19) that compare our product to MAXAR imagery, similar to Fig. 4 and Fig. S10 (Now S12).

Other comments:

1. In line 223 is not clear what are "burned and paired control sites".

   We agree this is confusing as we say paired on this line, but we are actually using a combination of unburned sites in our field database as well as 'paired' sites, which were not yet described. We have edited this paragraph to clarify. Paired sites themselves are described on lines 233-236, although we no longer use that terminology to describe them.

2. Line 387 and 389: dNBR is repeated.

   We have removed the second dNBR listing on line 392.

3. Line 475: how was the 3% standard deviation value obtained? A normal distribution was fitted to the data? The same for the 20% value in the next line; how was it obtained?

   Good questions. The 3% standard deviation was based on an original literature review conducted as part of Rogers et al. (2014). We added this citation in the text (Line 476). The 20% standard deviation for aboveground consumption was similarly based on past work (Walker et al., 2018; cited in the text), and comes from an expert interpretation in the field.

4. Line 503: I think it should be "We quantified uncertainty in our predictions **in** three ways", instated of ""We quantified uncertainty in our predictions three ways". This was the only grammar issue I found, but my mother tongue is Portuguese.

   We agree that the suggested sentence is better, and we have replaced it on line 504.

5. Line 549: "substantially" does not make sense looking to results (Fig. 3).

   We agree the term 'substantially' can be left up to interpretation, and we have removed the term on line 550.

6. Line 554: GFED4 is not in Fig. 3.

   We agree, the way we previously structured this is confusing. We have changed the sentence so that the figure references and areas are placed next to the text referenced.  Lines 555-557.

7. 4: f) e g) fire perimeters look the same.

   In this particular fire it is true that the Canadian Fire Database and the National Burned Area Composite are the same.  Only in some fires are they different.  We removed panel g) due to this, but the difference can clearly be seen in Figures S16-S19 on panels e and g.

8. Line 789: should be Figure S22 instead of Figure 22.

   This is correct, it should be Figure S22.  We have now corrected on line 792, the figure is now Fig. S32.

9. Line 850: for these kind of data with very large inter-annual variability, with some years that are strong outliers, it is better to use non-parametric methods to assess the trends, such as the Theil-Sen slope, which is very robust to outliers (significance is given by the associated test of Mann-Kendell).

   We agree Theil-Sen is more appropriate for this data than OLS.  We have replaced the confidence bounds and p-values to represent Theil-Sen slopes and significance from Mann-Kendall in Figures S36 and S37.  We have added text to the caption of each figure specifying this.

10. This is a suggestion since already many analyses were done in this work (but may be used in future research): it would be very ingesting to have maps of the trends in burned area and emissions. I mean to run spatiotemporal trends for Alaska and Canada and for the 19 years at the pixel level, done in a contextual way to account for the effect of the neighboring pixels. Some examples: https://link.springer.com/article/10.1007/s10113-018-1415-6, https://journals.plos.org/plosone/article?id=10.1371/journal.pone.0150663

We also agree it is instructive to assess temporal trends, which is why we included Figures 2 and S37. We do argue, however, that using 19 years to assess burned area trends in areas driven by long-interval, high severity fires such as Canada and Alaska is problematic, particularly at regional scales and smaller. Hence, we include only the trends at the continental scale.

**Citation**: https://doi.org/10.5194/egusphere-2022-364-RC1

---

## Author Comment (AC2)

In this work the authors evaluated the burned area and carbon emissions across Canada and Alaska over a 19-year period, using remote sensing data, field observation and modeling. The paper is well structured and clear, especially the introduction, results and discussion. However, the methodology could be resumed with flowcharts or some information added as supplementary material and the discussion could explore more perspectives and applications from the ABoVe-FED.

We thank the reviewer for their helpful comments. We have addressed each and believe the manuscript is stronger as a result. Original comments are given below, and our responses are in blue text.

Specific comments:

1. Line 149: The meaning of ABoVE domain is not cited early.

   We have added the meaning before the acronym on line 157.

2. Lines 175-180: Fig 1: The colors in the legend can be resumed by parenthesis instead of repeating "is shown in.."

   We agree it is easier to read in this fashion, and we have corrected the caption on Figure 1 on lines 184 -186.

3. In this figure its would interesting to show the final land cover map (for i.e. the last year of study, 2019)

   We added the land cover classifications as Figure S1.

4. Line 187: What did you mean by "burn depth"?

   We believe a detailed discussion of the methodologies of how burn depth and combustion are measured is beyond the scope of our methodology, and is suitably described in Rogers et al., 2014, Walker et al., 2018, and Dieleman et al., 2020, among others. Burn depth in this case refers to the vertical depth of burning in the soil organic column, which is directly related to carbon combustion. We have directed the reader to methodologies on line 192-194.

5. Lines of burned area mapping approach could be more resumed in a flowchart, for example.

   We agree with the authors comment and we have added a two flow charts, one for burned area (Figure S4) and one for combustion/burn depth (Figure S8) in the revisions which we believe will help readers follow the methodology.

6. Figure 8: add how much is "low fire occurrence"

   Of the total burned area less than 2% burns in these months across all years. We have edited the figure caption on lines 808-809.